# Machine learning the dimension of a Fano variety

Tom Coates [1], Alexander M. Kasprzyk [2] & Sara Veneziale [1] ✉

Fano varieties are basic building blocks in geometry – they are 'atomic pieces' of mathematical shapes. Recent progress in the classification of Fano varieties involves analysing an invariant called the quantum period. This is a sequence of integers which gives a numerical fingerprint for a Fano variety. It is conjectured that a Fano variety is uniquely determined by its quantum period. If this is true, one should be able to recover geometric properties of a Fano variety directly from its quantum period. We apply machine learning to the question: does the quantum period of $X$ know the dimension of $X$? Note that there is as yet no theoretical understanding of this. We show that a simple feed-forward neural network can determine the dimension of $X$ with 98% accuracy. Building on this, we establish rigorous asymptotics for the quantum periods of a class of Fano varieties. These asymptotics determine the dimension of $X$ from its quantum period. Our results demonstrate that machine learning can pick out structure from complex mathematical data in situations where we lack theoretical understanding. They also give positive evidence for the conjecture that the quantum period of a Fano variety determines that variety.

Algebraic geometry describes shapes as the solution sets of systems of polynomial equations, and manipulates or analyses a shape $X$ by manipulating or analysing the equations that define $X$. This interplay between algebra and geometry has applications across mathematics and science; see e.g., refs. 1–4. Shapes defined by polynomial equations are called *algebraic varieties*. Fano varieties are a key class of algebraic varieties. They are, in a precise sense, atomic pieces of mathematical shapes[5,6]. Fano varieties also play an essential role in string theory. They provide, through their 'anticanonical sections', the main construction of the Calabi–Yau manifolds which give geometric models of spacetime[7–9].

The classification of Fano varieties is a long-standing open problem. The only one-dimensional example is a line; this is classical. The ten smooth two-dimensional Fano varieties were found by del Pezzo in the 1880s[10]. The classification of smooth Fano varieties in dimension three was a triumph of 20th century mathematics: it combines work by Fano in the 1930s, Iskovskikh in the 1970s, and Mori–Mukai in the 1980s[11–16]. Beyond this, little is known, particularly for the important case of Fano varieties that are not smooth.

A new approach to Fano classification centres around a set of ideas from string theory called Mirror Symmetry[17–20]. From this perspective, the key invariant of a Fano variety is its *regularised quantum period*[21]

$$\widehat{G}_X(t) = \sum_{d=0}^{\infty} c_d t^d \tag{1}$$

This is a power series with coefficients $c_0 = 1$, $c_1 = 0$, and $c_d = r_d d!$, where $r_d$ is a certain Gromov–Witten invariant of $X$. Intuitively speaking, $r_d$ is the number of rational curves in $X$ of degree $d$ that pass through a fixed generic point and have a certain constraint on their complex structure. In general $r_d$ can be a rational number, because curves with a symmetry group of order $k$ are counted with weight $1/k$, but in all known cases the coefficients $c_d$ in (1) are integers.

It is expected that the regularised quantum period $\widehat{G}_X$ uniquely determines $X$. This is true (and proven) for smooth Fano varieties in low dimensions, but is unknown in dimensions four and higher, and for Fano varieties that are not smooth.

[1]Department of Mathematics, Imperial College London, London, UK. [2]School of Mathematical Sciences, University of Nottingham, Nottingham, UK. ✉e-mail: s.veneziale21@imperial.ac.uk

In this paper we will treat the regularised quantum period as a numerical signature for the Fano variety $X$, given by the sequence of integers $(c_0, c_1, \ldots)$. A priori this looks like an infinite amount of data, but in fact there is a differential operator $L$ such that $L\widehat{G}_X \equiv 0$; see e.g., [ref. 21, Theorem 4.3]. This gives a recurrence relation that determines all of the coefficients $c_d$ from the first few terms, so the regularised quantum period $\widehat{G}_X$ contains only a finite amount of information. Encoding a Fano variety $X$ by a vector in $\mathbb{Z}^{m+1}$ given by finitely many coefficients $(c_0, c_1, \ldots, c_m)$ of the regularised quantum period allows us to investigate questions about Fano varieties using machine learning.

In this paper, we ask whether the regularised quantum period of a Fano variety $X$ knows the dimension of $X$. There is currently no viable theoretical approach to this question. Instead, we use machine learning methods applied to a large dataset to argue that the answer is probably yes, and then prove that the answer is yes for toric Fano varieties of low Picard rank. The use of machine learning was essential to the formulation of our rigorous results (Theorems 5 and 6 below). This work is, therefore, proof-of-concept for a larger programme, demonstrating that machine learning can uncover previously unknown structure in complex mathematical datasets. Thus, the Data Revolution, which has had such impact across the rest of science, also brings important new insights to pure mathematics[22–27]. This is particularly true for large-scale classification questions, e.g., refs. [28–32], where these methods can potentially reveal both the classification itself and structural relationships within it.

## Results

### Algebraic varieties can be smooth or have singularities

Depending on their equations, algebraic varieties can be smooth (as in Fig. 1a) or have singularities (as in Fig. 1b). In this paper, we consider algebraic varieties over the complex numbers. The equations in Fig. 1a, b, therefore, define complex surfaces; however, for ease of visualisation, we have plotted only the points on these surfaces with co-ordinates that are real numbers.

Most of the algebraic varieties that we consider below will be singular, but they all have a class of singularities called *terminal quotient singularities*. This is the most natural class of singularities to allow from the point of view of Fano classification[6]. Terminal quotient singularities are very mild; indeed, in dimensions one and two, an algebraic variety has terminal quotient singularities if and only if it is smooth.

### The Fano varieties that we consider

The fundamental example of a Fano variety is projective space $\mathbb{P}^{N-1}$. This is a quotient of $\mathbb{C}^N \setminus \{0\}$ by the group $\mathbb{C}^\times$, where the action of

$\lambda \in \mathbb{C}^\times$ identifies the points $(x_1, x_2, \ldots, x_N)$ and $(\lambda x_1, \lambda x_2, \ldots, \lambda x_N)$. The resulting algebraic variety is smooth and has dimension $N-1$. We will consider generalisations of projective spaces called *weighted projective spaces* and *toric varieties of Picard rank two*. A detailed introduction to these spaces is given in the Supplementary Notes.

To define a weighted projective space, choose positive integers $a_1, a_2, \ldots, a_N$ such that any subset of size $N-1$ has no common factor, and consider

$$\mathbb{P}(a_1, a_2, \ldots, a_N) = (\mathbb{C}^N \setminus \{0\})/\mathbb{C}^\times$$

where the action of $\lambda \in \mathbb{C}^\times$ identifies the points

$$(x_1, x_2, \ldots, x_N) \quad \text{and} \quad (\lambda^{a_1} x_1, \lambda^{a_2} x_2, \ldots, \lambda^{a_N} x_N)$$

in $\mathbb{C}^N \setminus \{0\}$. The quotient $\mathbb{P}(a_1, a_2, \ldots, a_N)$ is an algebraic variety of dimension $N-1$. A general point of $\mathbb{P}(a_1, a_2, \ldots, a_N)$ is smooth, but there can be singular points. Indeed, a weighted projective space $\mathbb{P}(a_1, a_2, \ldots, a_N)$ is smooth if and only if $a_i = 1$ for all $i$, that is, if and only if it is a projective space.

To define a toric variety of Picard rank two, choose a matrix

$$\begin{pmatrix} a_1 & a_2 & \cdots & a_N \\ b_1 & b_2 & \cdots & b_N \end{pmatrix} \tag{2}$$

with non-negative integer entries and no zero columns. This defines an action of $\mathbb{C}^\times \times \mathbb{C}^\times$ on $\mathbb{C}^N$, where $(\lambda, \mu) \in \mathbb{C}^\times \times \mathbb{C}^\times$ identifies the points

$$(x_1, x_2, \ldots, x_N) \quad \text{and} \quad (\lambda^{a_1} \mu^{b_1} x_1, \lambda^{a_2} \mu^{b_2} x_2, \ldots, \lambda^{a_N} \mu^{b_N} x_N)$$

in $\mathbb{C}^N$. Set $a = a_1 + a_2 + \cdots + a_N$ and $b = b_1 + b_2 + \cdots + b_N$, and suppose that $(a, b)$ is not a scalar multiple of $(a_i, b_i)$ for any $i$. This determines linear subspaces

$$\begin{aligned} S_+ &= \{(x_1, x_2, \ldots, x_N) \mid x_i = 0 \text{ if } b_i/a_i < b/a\} \\ S_- &= \{(x_1, x_2, \ldots, x_N) \mid x_i = 0 \text{ if } b_i/a_i > b/a\} \end{aligned}$$

of $\mathbb{C}^N$, and we consider the quotient

$$X = (\mathbb{C}^N \setminus S)/(\mathbb{C}^\times \times \mathbb{C}^\times) \tag{3}$$

where $S = S_+ \cup S_-$. The quotient $X$ is an algebraic variety of dimension $N-2$ and second Betti number $b_2(X) \leq 2$. If, as we assume henceforth, the subspaces $S_+$ and $S_-$ both have dimension two or more then

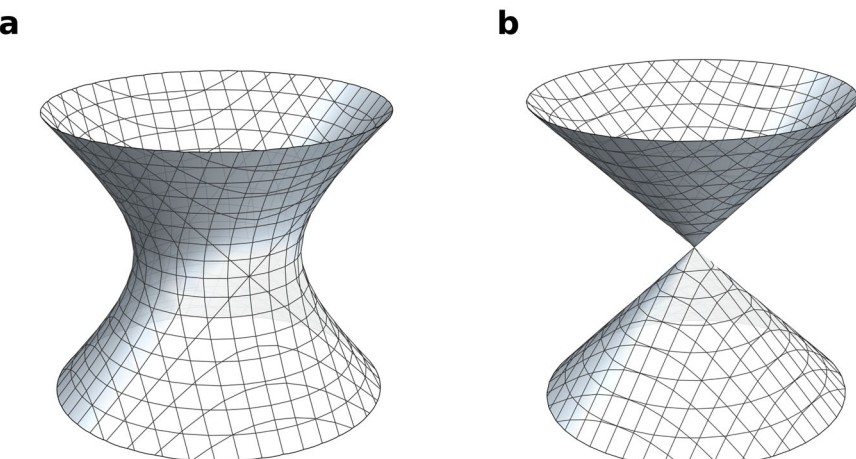

**a**  **b**

**Fig. 1 | Algebraic varieties and their equations. a** A smooth example ($x^2 + y^2 = z^2 + 1$); **b** An example with a singular point ($x^2 + y^2 = z^2$).

$b_2(X) = 2$, and thus $X$ has Picard rank two. In general $X$ will have singular points, the precise form of which is determined by the weights in (2).

There are closed formulas for the regularised quantum period of weighted projective spaces and toric varieties[33]. We have

$$\widehat{G}_{\mathbb{P}}(t) = \sum_{k=0}^{\infty} \frac{(ak)!}{(a_1 k)!(a_2 k)! \cdots (a_N k)!} t^{ak} \qquad (4)$$

where $\mathbb{P} = \mathbb{P}(a_1, \ldots, a_N)$ and $a = a_1 + a_2 + \cdots + a_N$, and

$$\widehat{G}_X(t) = \sum_{(k,l) \in \mathbb{Z}^2 \cap C} \frac{(ak + bl)!}{(a_1 k + b_1 l)! \cdots (a_N k + b_N l)!} t^{ak+bl} \qquad (5)$$

where the weights for $X$ are as in (2), and $C$ is the cone in $\mathbb{R}^2$ defined by the equations $a_i x + b_i y \geq 0$, $i \in \{1, 2, \ldots, N\}$. Formula (4) implies that, for weighted projective spaces, the coefficient $c_d$ from (1) is zero unless $d$ is divisible by $a$. Formula (5) implies that, for toric varieties of Picard rank two, $c_d = 0$ unless $d$ is divisible by $\gcd\{a, b\}$.

## Data generation: weighted projective spaces
The following result characterises weighted projective spaces with terminal quotient singularities; this is [ref. 34, Proposition 2.3].

**Proposition 1.** Let $X = \mathbb{P}(a_1, a_2, \ldots, a_N)$ be a weighted projective space of dimension at least three. Then $X$ has terminal quotient singularities if and only if

$$\sum_{i=1}^{N} \{ka_i/a\} \in \{2, \ldots, N-2\}$$

for each $k \in \{2, \ldots, a-2\}$. Here $a = a_1 + a_2 + \cdots + a_N$ and $\{q\}$ denotes the fractional part $q - \lfloor q \rfloor$ of $q \in \mathbb{Q}$.

A simpler necessary condition is given by [ref. 35, Theorem 3.5]:

**Proposition 2.** Let $X = \mathbb{P}(a_1, a_2, \ldots, a_N)$ be a weighted projective space of dimension at least two, with weights ordered $a_1 \leq a_2 \leq \ldots \leq a_N$. If $X$ has terminal quotient singularities then $a_i/a < 1/(N - i + 2)$ for each $i \in \{3, \ldots, N\}$.

Weighted projective spaces with terminal quotient singularities have been classified in dimensions up to four[34,36]. Classifications in higher dimensions are hindered by the lack of an effective upper bound on $a$.

We randomly generated 150,000 distinct weighted projective spaces with terminal quotient singularities, and with dimension up to 10, as follows. We generated random sequences of weights $a_1 \leq a_2 \leq \ldots \leq a_N$ with $a_N \leq 10N$ and discarded them if they failed to satisfy any one of the following:

1. for each $i \in \{1, \ldots, N\}$, $\gcd\{a_1, \ldots, \widehat{a}_i, \ldots, a_N\} = 1$, where $\widehat{a}_i$ indicates that $a_i$ is omitted;
2. $a_i/a < 1/(N - i + 2)$ for each $i \in \{3, \ldots, N\}$;
3. $\sum_{i=1}^{N} \{ka_i/a\} \in \{2, \ldots, N-2\}$ for each $k \in \{2, \ldots, a-2\}$.

Condition 1 here was part of our definition of weighted projective spaces above; it ensures that the set of singular points in $\mathbb{P}(a_1, a_2, \ldots, a_N)$ has dimension at most $N-2$, and also that weighted projective spaces are isomorphic as algebraic varieties if and only if they have the same weights. Condition 2 is from Proposition 2; it efficiently rules out many non-terminal examples. Condition 3 is the necessary and sufficient condition from Proposition 1. We then deduplicated the sequences. The resulting sample sizes are summarised in Table 1.

## Data generation: toric varieties
Deduplicating randomly-generated toric varieties of Picard rank two is harder than deduplicating randomly-generated weighted projective spaces, because different weight matrices in (2) can give rise to the same toric variety. Toric varieties are uniquely determined, up to isomorphism, by a combinatorial object called a *fan*[37]. A fan is a collection of cones, and one can determine the singularities of a toric variety $X$ from the geometry of the cones in the corresponding fan.

We randomly generated 200,000 distinct toric varieties of Picard rank two with terminal quotient singularities, and with dimension up to 10, as follows. We randomly generated weight matrices, as in (2), such that $0 \leq a_i, b_j \leq 5$. We then discarded the weight matrix if any column was zero, and otherwise formed the corresponding fan $F$. We discarded the weight matrix unless:

1. $F$ had $N$ rays;
2. each cone in $F$ was simplicial (i.e., has number of rays equal to its dimension);
3. the convex hull of the primitive generators of the rays of $F$ contained no lattice points other than the rays and the origin.

Conditions 1 and 2 together guarantee that $X$ has Picard rank two, and are equivalent to the conditions on the weight matrix in (2) given in our definition. Conditions 2 and 3 guarantee that $X$ has terminal quotient singularities. We then deduplicated the weight matrices according to the isomorphism type of $F$, by putting $F$ in normal form[38,39]. See Table 1 for a summary of the dataset.

## Data analysis: weighted projective spaces
We computed an initial segment $(c_0, c_1, \ldots, c_m)$ of the regularised quantum period for all the examples in the sample of 150,000 terminal weighted projective spaces, with $m \approx 100,000$. The non-zero coefficients $c_d$ appeared to grow exponentially with $d$, and so we considered

## Table 1 | The distribution by dimension in our datasets

| Weighted projective spaces | | | Rank-two toric varieties | | |
|---|---|---|---|---|---|
| Dimension | Sample size | Percentage | Dimension | Sample size | Percentage |
| 1 | 1 | 0.001 | | | |
| 2 | 1 | 0.001 | 2 | 2 | 0.001 |
| 3 | 7 | 0.005 | 3 | 17 | 0.009 |
| 4 | 8936 | 5.957 | 4 | 758 | 0.379 |
| 5 | 23,584 | 15.723 | 5 | 6050 | 3.025 |
| 6 | 23,640 | 15.760 | 6 | 19,690 | 9.845 |
| 7 | 23,700 | 15.800 | 7 | 35,395 | 17.698 |
| 8 | 23,469 | 15.646 | 8 | 42,866 | 21.433 |
| 9 | 23,225 | 15.483 | 9 | 47,206 | 23.603 |
| 10 | 23,437 | 15.625 | 10 | 48,016 | 24.008 |
| Total | 150,000 | | Total | 200,000 | |

**a**

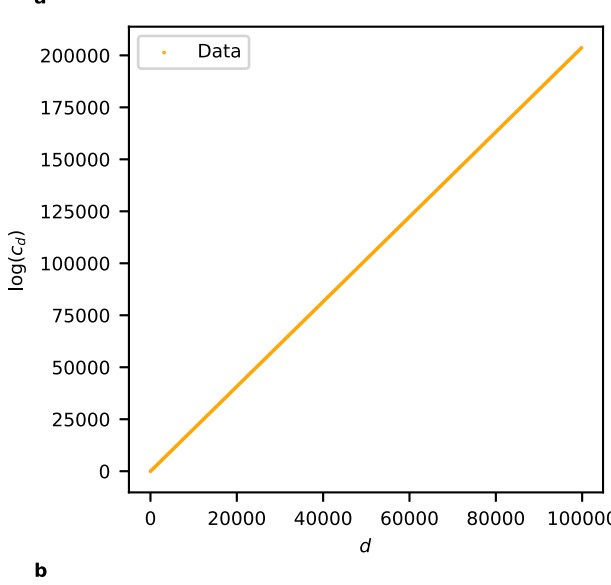

**b**

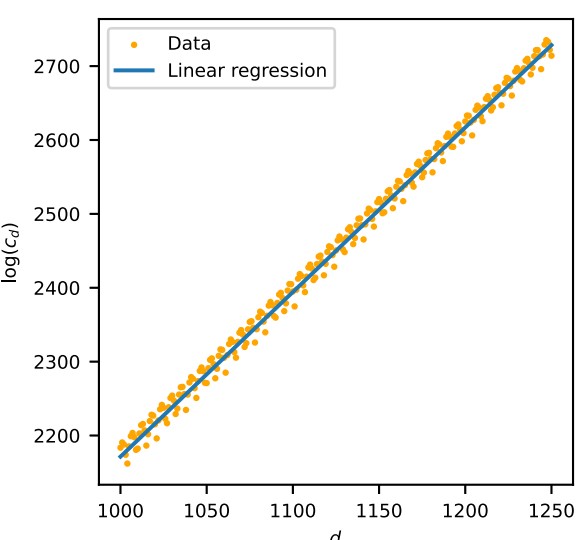

**Fig. 2 | The logarithm of the non-zero period coefficients $c_d$: a** for a typical weighted projective space ($\mathbb{P}$(5,5,11,23,28,29,33,44,66,76)); **b** for the toric variety of Picard rank two from Example 3.

$\{\log c_d\}_{d \in S}$ where $S = \{d \in \mathbb{Z}_{\geq 0} | c_d \neq 0\}$. To reduce dimension, we fitted a linear model to the set $\{(d, \log c_d) | d \in S\}$ and used the slope and intercept of this model as features; see Fig. 2a for a typical example. Plotting the slope against the $y$-intercept and colouring datapoints according to the dimension we obtain Fig. 3a: note the clear separation by dimension. A Support Vector Machine (SVM) trained on 10% of the slope and $y$-intercept data predicted the dimension of the weighted projective space with an accuracy of 99.99%. Full details are given in the Supplementary Methods.

### Data analysis: toric varieties
As before, the non-zero coefficients $c_d$ appeared to grow exponentially with $d$, so we fitted a linear model to the set $\{(d, \log c_d) | d \in S\}$ where $S = \{d \in \mathbb{Z}_{\geq 0} | c_d \neq 0\}$. We used the slope and intercept of this linear model as features.

**Example 3.** In Fig. 2b, we plot a typical example: the logarithm of the regularised quantum period sequence for the nine-dimensional toric

variety with weight matrix

$$\begin{pmatrix} 1 & 2 & 5 & 3 & 3 & 3 & 0 & 0 & 0 & 0 & 0 \\ 0 & 0 & 0 & 3 & 4 & 4 & 1 & 2 & 2 & 3 & 4 \end{pmatrix}$$

along with the linear approximation. We see a periodic deviation from the linear approximation; the magnitude of this deviation decreases as $d$ increases (not shown).

To reduce computational costs, we computed pairs $(d, \log c_d)$ for $1000 \leq d \leq 20{,}000$ by sampling every 100th term. We discarded the beginning of the period sequence because of the noise it introduces to the linear regression. In cases where the sampled coefficient $c_d$ is zero, we considered instead the next non-zero coefficient. The resulting plot of slope against $y$-intercept, with datapoints coloured according to dimension, is shown in Fig. 3b.

We analysed the standard errors for the slope and $y$-intercept of the linear model. The standard errors for the slope are small compared to the range of slopes but, in many cases, the standard error $s_{\text{int}}$ for the $y$-intercept is relatively large. As Fig. 4 illustrates, discarding data points where the standard error $s_{\text{int}}$ for the $y$-intercept exceeds some threshold reduces apparent noise. This suggests that the underlying structure is being obscured by inaccuracies in the linear regression caused by oscillatory behaviour in the initial terms of the quantum period sequence; these inaccuracies are concentrated in the $y$-intercept of the linear model. Note that restricting attention to those data points where $s_{\text{int}}$ is small also greatly decreases the range of $y$-intercepts that occur. As Example 4 and Fig. 5 suggest, this reflects both transient oscillatory behaviour and also the presence of a subleading term in the asymptotics of $\log c_d$ which is missing from our feature set. We discuss this further below.

**Example 4.** Consider the toric variety with Picard rank two and weight matrix

$$\begin{pmatrix} 1 & 10 & 5 & 13 & 8 & 12 & 0 \\ 0 & 0 & 3 & 8 & 5 & 14 & 1 \end{pmatrix}$$

This is one of the outliers in Fig. 3b. The toric variety is five-dimensional, and has slope 1.637 and $y$-intercept −62.64. The standard errors are $4.246 \times 10^{-4}$ for the slope and 5.021 for the $y$-intercept. We computed the first 40 000 coefficients $c_d$ in (1). As Fig. 5 shows, as $d$ increases the $y$-intercept of the linear model increases to −28.96 and $s_{\text{int}}$ decreases to 0.7877. At the same time, the slope of the linear model remains more or less unchanged, decreasing to 1.635. This supports the idea that computing (many) more coefficients $c_d$ would significantly reduce noise in Fig. 3b. In this example, even 40,000 coefficients may not be enough.

Computing many more coefficients $c_d$ across the whole dataset would require impractical amounts of computation time. In the example above, which is typical in this regard, increasing the number of coefficients computed from 20,000 to 40,000 increased the computation time by a factor of more than 10. Instead we restrict to those toric varieties of Picard rank two such that the $y$-intercept standard error $s_{\text{int}}$ is less than 0.3; this retains 67,443 of the 200,000 datapoints. We used 70% of the slope and $y$-intercept data in the restricted dataset for model training, and the rest for validation. An SVM model predicted the dimension of the toric variety with an accuracy of 87.7%, and a Random Forest Classifier (RFC) predicted the dimension with an accuracy of 88.6%.

### Neural networks
Neural networks do not handle unbalanced datasets well. Therefore, we removed the toric varieties of dimensions 3, 4, and 5 from our data, leaving 61,164 toric varieties of Picard rank two with terminal quotient

**a**

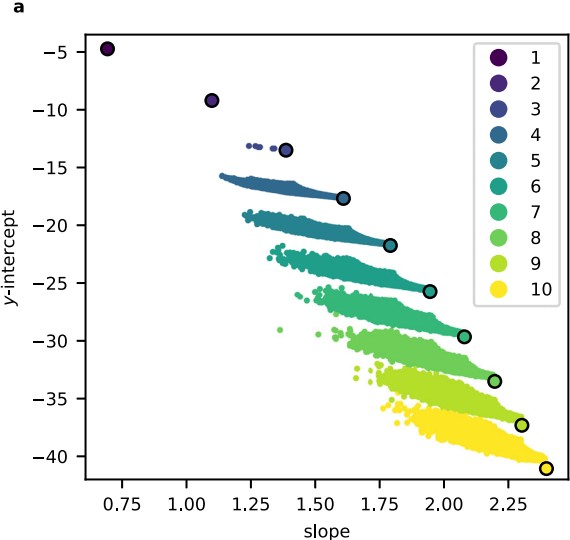
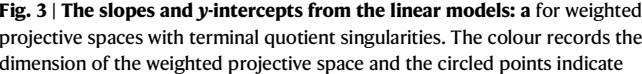

**b**

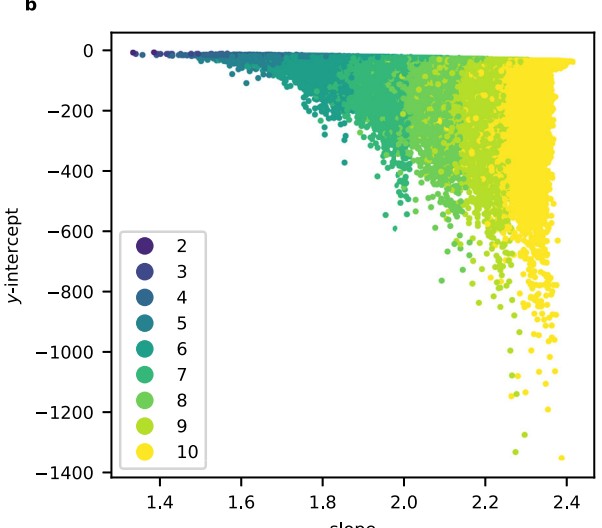

**Fig. 3 | The slopes and *y*-intercepts from the linear models: a** for weighted projective spaces with terminal quotient singularities. The colour records the dimension of the weighted projective space and the circled points indicate

projective spaces. **b** for toric varieties of Picard rank two with terminal quotient singularities. The colour records the dimension of the toric variety.

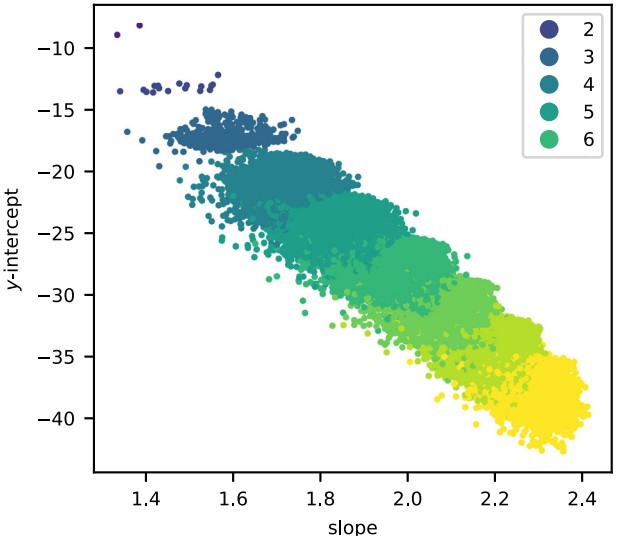

**Fig. 4 | The slopes and *y*-intercepts from the linear model.** This is as in Fig. 3b, but plotting only data points for which the standard error $s_{int}$ for the *y*-intercept satisfies $s_{int} < 0.3$. The colour records the dimension of the toric variety.

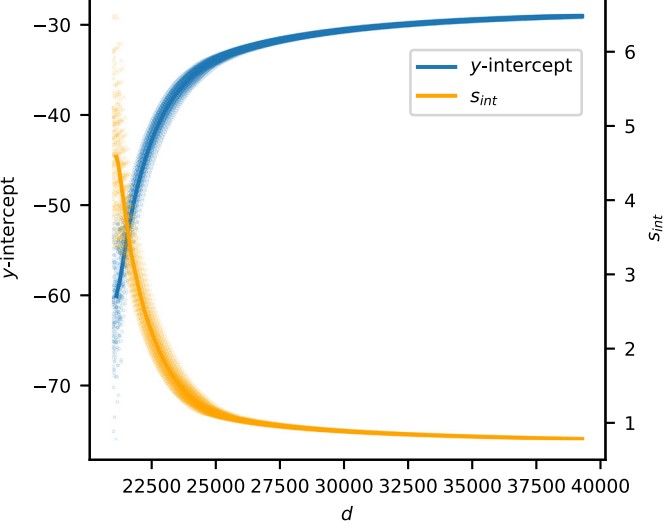

**Fig. 5 | Variation as we move deeper into the period sequence.** The *y*-intercept and its standard error $s_{int}$ for the toric variety from Example 4, as computed from pairs $(k, \log c_k)$ such that $d - 20{,}000 \le k \le d$ by sampling every 100th term. We also show LOWESS-smoothed trend lines.

singularities and $s_{int} < 0.3$. This dataset is approximately balanced by dimension.

A Multilayer Perceptron (MLP) with three hidden layers of sizes (10, 30, 10) using the slope and intercept as features predicted the dimension with 89.0% accuracy. Since the slope and intercept give good control over $\log c_d$ for $d \gg 0$, but not for small $d$, it is likely that the coefficients $c_d$ with $d$ small contain extra information that the slope and intercept do not see. Supplementing the feature set by including the first 100 coefficients $c_d$ as well as the slope and intercept increased the accuracy of the prediction to 97.7%. Full details can be found in the Supplementary Methods.

## From machine learning to rigorous analysis
Elementary "out of the box" models (SVM, RFC, and MLP) trained on the slope and intercept data alone already gave a highly accurate prediction for the dimension. Furthermore, even for the many-feature

MLP, which was the most accurate, sensitivity analysis using SHAP values[40] showed that the slope and intercept were substantially more important to the prediction than any of the coefficients $c_d$: see Fig. 6. This suggested that the dimension of $X$ might be visible from a rigorous estimate of the growth rate of $\log c_d$.

In the Methods section, we establish asymptotic results for the regularised quantum period of toric varieties with low Picard rank, as follows. These results apply to any weighted projective space or toric variety of Picard rank two: they do not require a terminality hypothesis. Note, in each case, the presence of a subleading logarithmic term in the asymptotics for $\log c_d$.

**Theorem 5.** Let $X$ denote the weighted projective space $\mathbb{P}(a_1, \ldots, a_N)$, so that the dimension of $X$ is $N - 1$. Let $c_d$ denote the coefficient of $t^d$ in the regularised quantum period $\widehat{G}_X(t)$ given in (4). Let $a = a_1 + \cdots + a_N$

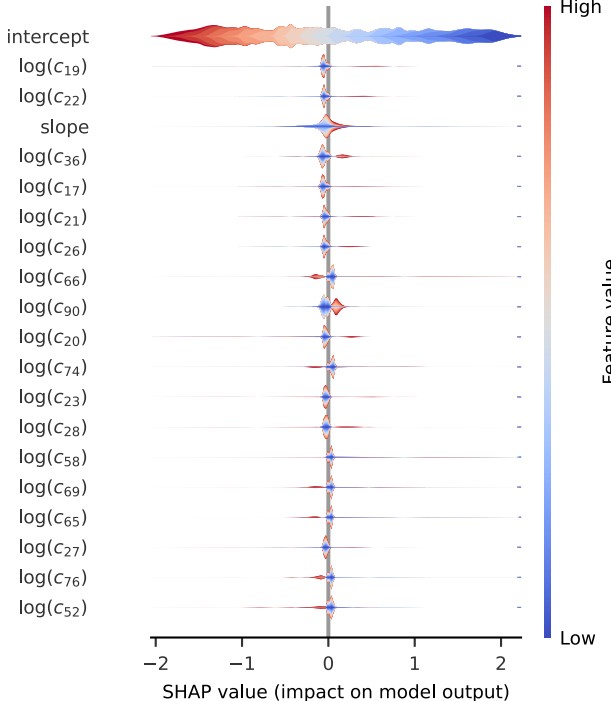

**Fig. 6 | Model sensitivity analysis using SHAP values.** The model is an MLP with three hidden layers of sizes (10,30,10) applied to toric varieties of Picard rank two with terminal quotient singularities. It is trained on the slope, $y$-intercept, and the first 100 coefficients $c_d$ as features, and predicts the dimension with 97.7% accuracy.

and $p_i = a_i/a$. Then $c_d = 0$ unless $d$ is divisible by $a$, and non-zero coefficients $c_d$ satisfy

$$\log c_d \sim Ad - \frac{\dim X}{2}\log d + B$$

as $d \to \infty$, where

$$A = -\sum_{i=1}^{N} p_i \log p_i$$

$$B = -\frac{\dim X}{2}\log(2\pi) - \frac{1}{2}\sum_{i=1}^{N} \log p_i$$

Note, although it plays no role in what follows, that $A$ is the Shannon entropy of the discrete random variable $Z$ with distribution $(p_1, p_2, ..., p_N)$, and that $B$ is a constant plus half the total self-information of $Z$.

**Theorem 6.** Let $X$ denote the toric variety of Picard rank two with weight matrix

$$\begin{pmatrix} a_1 & a_2 & a_3 & \cdots & a_N \\ b_1 & b_2 & b_3 & \cdots & b_N \end{pmatrix}$$

so that the dimension of $X$ is $N - 2$. Let $a = a_1 + \cdots + a_N$, $b = b_1 + \cdots + b_N$, and $\ell = \gcd\{a, b\}$. Let $[\mu : \nu] \in \mathbb{P}^1$ be the unique root of the homogeneous polynomial

$$\prod_{i=1}^{N}(a_i\mu + b_i\nu)^{a_ib} - \prod_{i=1}^{N}(a_i\mu + b_i\nu)^{b_ia}$$

such that $a_i\mu + b_i\nu \geq 0$ for all $i \in \{1, 2, ..., N\}$, and set

$$p_i = \frac{\mu a_i + \nu b_i}{\mu a + \nu b}$$

Let $c_d$ denote the coefficient of $t^d$ in the regularised quantum period $\widehat{G}_X(t)$ given in (5). Then non-zero coefficients $c_d$ satisfy

$$\log c_d \sim Ad - \frac{\dim X}{2}\log d + B$$

as $d \to \infty$, where

$$A = -\sum_{i=1}^{N} p_i \log p_i$$

$$B = -\frac{\dim X}{2}\log(2\pi) - \frac{1}{2}\sum_{i=1}^{N} \log p_i - \frac{1}{2}\log\left(\sum_{i=1}^{N}\frac{(a_ib - b_ia)^2}{\ell^2 p_i}\right)$$

Theorem 5 is a straightforward application of Stirling's formula. Theorem 6 is more involved, and relies on a Central Limit-type theorem that generalises the De Moivre–Laplace theorem.

## Theoretical analysis

The asymptotics in Theorems 5 and 6 imply that, for $X$ a weighted projective space or toric variety of Picard rank two, the quantum period determines the dimension of $X$. Let us revisit the clustering analysis from this perspective. Recall the asymptotic expression $\log c_d \sim Ad - \frac{\dim X}{2}\log d + B$ and the formulae for $A$ and $B$ from Theorem 5. Figure 7a shows the values of $A$ and $B$ for a sample of weighted projective spaces, coloured by dimension. Note the clusters, which overlap. Broadly speaking, the values of $B$ increase as the dimension of the weighted projective space increases, whereas in Fig. 3a, the $y$-intercepts decrease as the dimension increases. This reflects the fact that we fitted a linear model to $\log c_d$, omitting the subleading $\log d$ term in the asymptotics. As Fig. 8 shows, the linear model assigns the omitted term to the $y$-intercept rather than the slope. The slope of the linear model is approximately equal to $A$. The $y$-intercept, however, differs from $B$ by a dimension-dependent factor. The omitted log term does not vary too much over the range of degrees ($d < 100,000$) that we considered, and has the effect of reducing the observed $y$-intercept from $B$ to approximately $B - \frac{9}{2}\dim X$, distorting the clusters slightly and translating them downwards by a dimension-dependent factor. This separates the clusters. We expect that the same mechanism applies in Picard rank two as well: see Fig. 7b.

We can show that each cluster in Fig. 7a is linearly bounded using constrained optimisation techniques. Consider, for example, the cluster for weighted projective spaces of dimension five, as in Fig. 9.

**Proposition 7.** Let $X$ be the five-dimensional weighted projective space $\mathbb{P}(a_1, ..., a_6)$, and let $A$, $B$ be as in Theorem 5. Then $B + \frac{5}{2}A \geq \frac{41}{8}$. If in addition $a_i \leq 25$ for all $i$ then $B + 5A \leq \frac{41}{40}$.

Fix a suitable $\theta \geq 0$ and consider

$$B + \theta A = -\frac{\dim X}{2}\log(2\pi) - \frac{1}{2}\sum_{i=1}^{N}\log p_i - \theta\sum_{i=1}^{N}p_i\log p_i$$

with $\dim X = N - 1 = 5$. Solving

$$\min(B + \theta A) \quad \text{subject to} \quad p_1 + \cdots + p_6 = 1$$
$$p_1, ..., p_6 \geq 0$$

on the five-simplex gives a linear lower bound for the cluster. This bound does not use terminality: it applies to any weighted projective space of dimension five. The expression $B + \theta A$ is unbounded above on

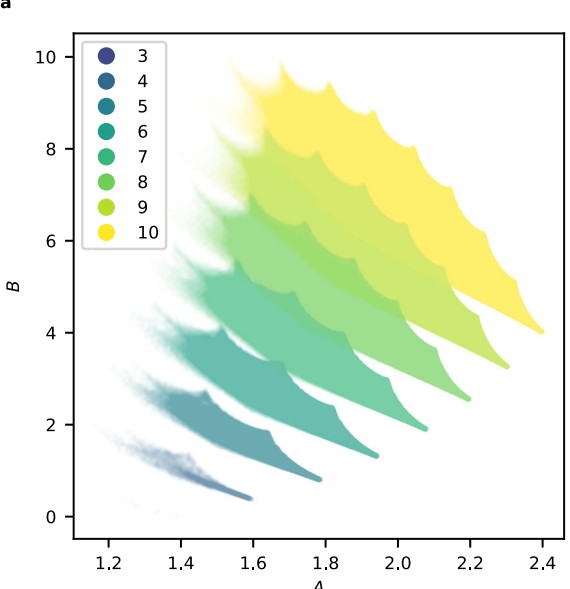

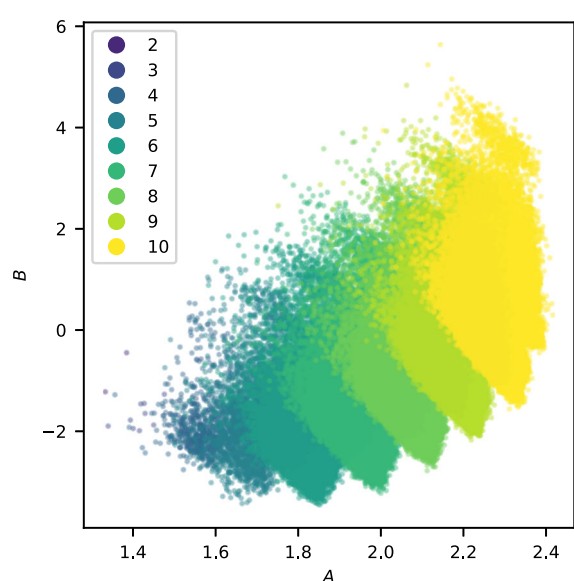

**Fig. 7 | The values of the asymptotic coefficients *A* and *B*: a** for all weighted projective spaces $\mathbb{P}(a_1,\ldots,a_N)$ with terminal quotient singularities and $a_i \leq 25$ for all $i$. The colour records the dimension of the weighted projective space. **b** for toric varieties of Picard rank two in our dataset. The colour records the dimension of the toric variety.

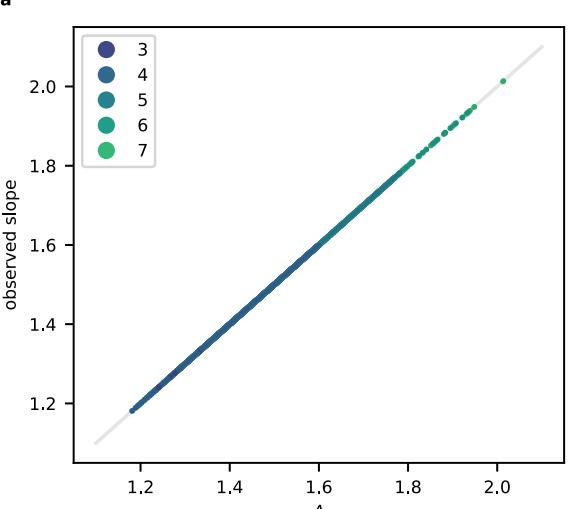

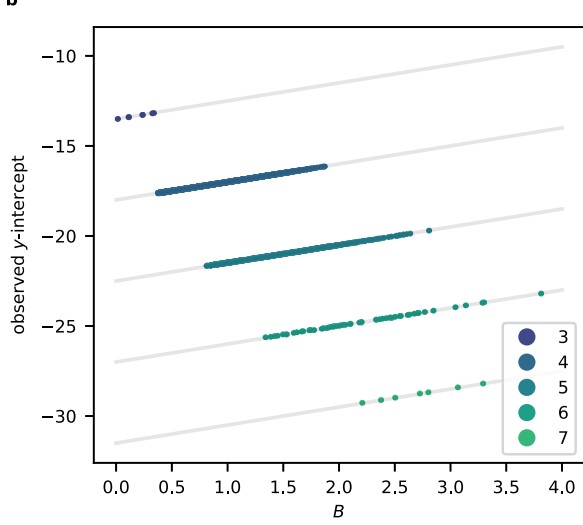

**Fig. 8 | For weighted projective spaces, the asymptotic coefficients *A* and *B* are closely related to the slope and *y*-intercept. a** Comparison between *A* and the slope from the linear model, for weighted projective spaces that occur in both Figs. 3a and 7, coloured by dimension. The line slope = *A* is indicated. **b** Comparison between *B* and the *y*-intercept from the linear model, for weighted projective spaces that occur in both Figs 3a and 7a, coloured by dimension. In each case, the line *y*-intercept = $B - \frac{9}{2}\dim X$ is shown.

the five-simplex (because *B* is) so we cannot obtain an upper bound this way. Instead, consider

$$\max(B + \theta A) \quad \text{subject to} \quad \begin{aligned} &p_1 + \cdots + p_6 = 1\\ &\epsilon \leq p_1 \leq p_2 \leq \cdots \leq p_6 \end{aligned}$$

for an appropriate small positive $\epsilon$, which we can take to be $1/a$ where $a$ is the maximum sum of the weights. For Fig. 9, for example, we can take $a = 124$, and in general, such an $a$ exists because there are only finitely many terminal weighted projective spaces. This gives a linear upper bound for the cluster.

The same methods yield linear bounds on each of the clusters in Fig. 7a. As the Figure shows, however, the clusters are not linearly separable.

## Discussion

We developed machine learning models that predict, with high accuracy, the dimension of a Fano variety from its regularised quantum period. These models apply to weighted projective spaces and toric varieties of Picard rank two with terminal quotient singularities. We then established rigorous asymptotics for the regularised quantum period of these Fano varieties. The form of the asymptotics implies that, in these cases, the regularised quantum period of a Fano variety *X* determines the dimension of *X*. The asymptotics also give a theoretical underpinning for the success of the machine learning models.

Perversely, because the series involved converge extremely slowly, reading the dimension of a Fano variety directly from the asymptotics of the regularised quantum period is not practical. For the

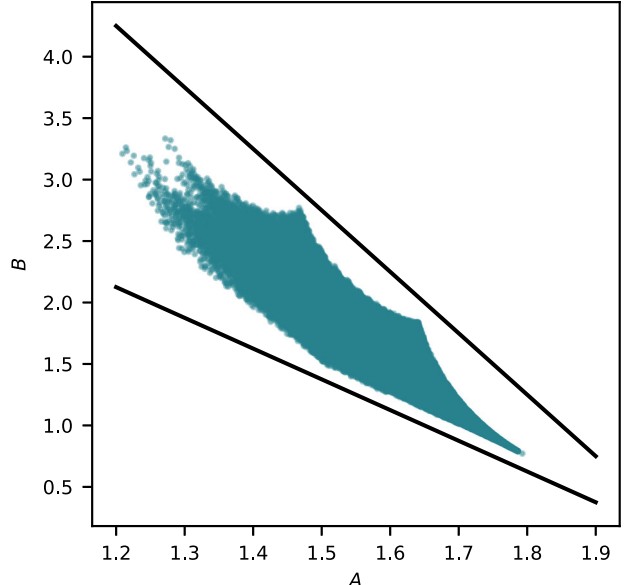

**Fig. 9 | Linear bounds for the cluster of five-dimensional weighted projective spaces in Fig. 7a.** The bounds are given by Proposition 7.

same reason, enhancing the feature set of our machine learning models by including a $\log d$ term in the linear regression results in less accurate predictions. So although the asymptotics in Theorems 5 and 6 determine the dimension in theory, in practice, the most effective way to determine the dimension of an unknown Fano variety from its quantum period is to apply a machine learning model.

The insights gained from machine learning were the key to our formulation of the rigorous results in Theorems 5 and 6. Indeed, it might be hard to discover these results without a machine learning approach. It is notable that the techniques in the proof of Theorem 6 – the identification of generating functions for Gromov–Witten invariants of toric varieties with certain hypergeometric functions – have been known since the late 1990s and have been studied by many experts in hypergeometric functions since then. For us, the essential step in the discovery of the results was the feature extraction that we performed as part of our ML pipeline.

This work demonstrates that machine learning can uncover previously unknown structure in complex mathematical data, and is a powerful tool for developing rigorous mathematical results; cf.[22]. It also provides evidence for a fundamental conjecture in the Fano classification programme[21]: that the regularised quantum period of a Fano variety determines that variety.

## Methods

In this section, we prove Theorem 5 and Theorem 6. The following result implies Theorem 5.

**Theorem 8.** Let $X$ denote the weighted projective space $\mathbb{P}(a_1, \ldots, a_N)$, so that the dimension of $X$ is $N-1$. Let $c_d$ denote the coefficient of $t^d$ in the regularised quantum period $\widehat{G}_X(t)$ given in (4). Let $a = a_1 + \ldots + a_N$. Then $c_d = 0$ unless $d$ is divisible by $a$, and

$$\log c_{ka} \sim ka \left[ \log a - \frac{1}{a} \sum_{i=1}^{N} a_i \log a_i \right] - \frac{\dim X}{2} \log(ka)$$
$$+ \frac{1 + \dim X}{2} \log a - \frac{\dim X}{2} \log(2\pi) - \frac{1}{2} \sum_{i=1}^{N} \log a_i$$

That is, non-zero coefficients $c_d$ satisfy

$$\log c_d \sim Ad - \frac{\dim X}{2} \log d + B$$

as $d \to \infty$, where

$$A = -\sum_{i=1}^{N} p_i \log p_i \quad B = -\frac{\dim X}{2} \log(2\pi) - \frac{1}{2} \sum_{i=1}^{N} \log p_i$$

and $p_i = a_i/a$.

**Proof.** Combine Stirling's formula

$$n! \sim \sqrt{2\pi n} \left( \frac{n}{e} \right)^n$$

with the closed formula (4) for $c_{ka}$. □

### Toric varieties of Picard rank 2

Consider a toric variety $X$ of Picard rank two and dimension $N-2$ with weight matrix

$$\begin{pmatrix} a_1 & a_2 & a_3 & \cdots & a_N \\ b_1 & b_2 & b_3 & \cdots & b_N \end{pmatrix}$$

as in (2). Let us move to more invariant notation, writing $\alpha_i$ for the linear form on $\mathbb{R}^2$ defined by the transpose of the $i$th column of the weight matrix, and $\alpha = \alpha_1 + \cdots + \alpha_N$. Eq. (5) becomes

$$\widehat{G}_X(t) = \sum_{k \in \mathbb{Z}^2 \cap C} \frac{(\alpha \cdot k)!}{\prod_{i=1}^{N} (\alpha_i \cdot k)!} t^{\alpha \cdot k}$$

where $C$ is the cone $C = \{x \in \mathbb{R}^2 | \alpha_i \cdot x \geq 0 \text{ for } i = 1, 2, \ldots, N\}$. As we will see, for $d \gg 0$ the coefficients

$$\frac{(\alpha \cdot k)!}{\prod_{i=1}^{N} (\alpha_i \cdot k)!} \quad \text{where } k \in \mathbb{Z}^2 \cap C \text{ and } \alpha \cdot k = d$$

are approximated by a rescaled Gaussian. We begin by finding the mean of that Gaussian, that is, by minimising

$$\prod_{i=1}^{N} (\alpha_i \cdot k)! \quad \text{where } k \in \mathbb{Z}^2 \cap C \text{ and } \alpha \cdot k = d.$$

For $k$ in the strict interior of $C$ with $\alpha \cdot k = d$, we have that

$$(\alpha_i \cdot k)! \sim \left( \frac{\alpha_i \cdot k}{e} \right)^{\alpha_i \cdot k}$$

as $d \to \infty$.

**Proposition 9.** The constrained optimisation problem

$$\min \prod_{i=1}^{N} (\alpha_i \cdot x)^{\alpha_i \cdot x} \quad \text{subject to} \begin{cases} x \in C \\ \alpha \cdot x = d \end{cases}$$

has a unique solution $x = x^\bullet$. Furthermore, setting $p_i = (\alpha_i \cdot x^\bullet)/(\alpha \cdot x^\bullet)$ we have that the monomial

$$\prod_{i=1}^{N} p_i^{\alpha_i \cdot k}$$

depends on $k \in \mathbb{Z}^2$ only via $\alpha \cdot k$.

**Proof.** Taking logarithms gives the equivalent problem

$$\min \sum_{i=1}^{N} (\alpha_i \cdot x) \log(\alpha_i \cdot x) \quad \text{subject to} \begin{cases} x \in C \\ \alpha \cdot x = d \end{cases} \tag{6}$$

The objective function $\sum_{i=1}^{N}(\alpha_i \cdot x)\log(\alpha_i \cdot x)$ here is the pullback to $\mathbb{R}^2$ of the function

$$f(x_1, \ldots, x_N) = \sum_{i=1}^{N} x_i \log x_i$$

along the linear embedding $\varphi : \mathbb{R}^2 \to \mathbb{R}^N$ given by $(\alpha_1, \ldots, \alpha_N)$. Note that $C$ is the preimage under $\varphi$ of the positive orthant $\mathbb{R}^N_+$, so we need to minimise $f$ on the intersection of the simplex $x_1 + \cdots + x_N = d$, $(x_1, \ldots, x_N) \in \mathbb{R}^N_+$ with the image of $\varphi$. The function $f$ is convex and decreases as we move away from the boundary of the simplex, so the minimisation problem in Eq. (6) has a unique solution $x^*$ and this lies in the strict interior of $C$. We can, therefore, find the minimum $x^*$ using the method of Lagrange multipliers, by solving

$$\sum_{i=1}^{N} \alpha_i \log(\alpha_i \cdot x) + \alpha = \lambda \alpha \tag{7}$$

for $\lambda \in \mathbb{R}$ and $x$ in the interior of $C$ with $\alpha \cdot x = d$. Thus

$$\sum_{i=1}^{N} \alpha_i \log(\alpha_i \cdot x^*) = (\lambda - 1)\alpha$$

and, evaluating on $k \in \mathbb{Z}^2$ and exponentiating, we see that

$$\prod_{i=1}^{N} (\alpha_i \cdot x^*)^{\alpha_i \cdot k}$$

depends only on $\alpha \cdot k$. The result follows. □

Given a solution $x^*$ to Eq. (7), any positive scalar multiple of $x^*$ also satisfies Eq. (7), with a different value of $\lambda$ and a different value of $d$. Thus the solutions $x^*$, as $d$ varies, lie on a half-line through the origin. The direction vector $[\mu : \nu] \in \mathbb{P}^1$ of this half-line is the unique solution to the system

$$\prod_{i=1}^{N} (a_i \mu + b_i \nu)^{a_i b} = \prod_{i=1}^{N} (a_i \mu + b_i \nu)^{b_i a}$$
$$\begin{pmatrix} \mu \\ \nu \end{pmatrix} \in C \tag{8}$$

Note that the first equation here is homogeneous in $\mu$ and $\nu$; it is equivalent to Eq. (7), by exponentiating and then eliminating $\lambda$. Any two solutions $x^*$, for different values of $d$, differ by rescaling, and the quantities $p_i$ in Proposition 9 are invariant under this rescaling. They also satisfy $p_1 + \cdots + p_N = 1$.

We use the following result, known in the literature as the "Local Theorem"[41], to approximate multinomial coefficients.

**Local Theorem.** For $p_1, \ldots, p_n \in [0, 1]$ such that $p_1 + \cdots + p_n = 1$, the ratio

$$d^{\frac{n-1}{2}} \binom{d}{k_1 \cdots k_n} \prod_{i=1}^{n} p_i^{k_i} : \frac{\exp(-\frac{1}{2}\sum_{i=1}^{n} q_i x_i^2)}{(2\pi)^{\frac{n-1}{2}} \sqrt{p_1 \cdots p_n}} \to 1$$

as $d \to \infty$, uniformly in all $k_i$'s, where

$$q_i = 1 - p_i \quad x_i = \frac{k_i - dp_i}{\sqrt{dp_i q_i}}$$

and the $x_i$ lie in bounded intervals.

Let $B_r$ denote the ball of radius $r$ about $x^* \in \mathbb{R}^2$. Fix $R > 0$. We apply the Local Theorem with $k_i = \alpha_i \cdot k$ and $p_i = (\alpha_i \cdot x^*)/(\alpha \cdot x^*)$, where $k \in \mathbb{Z}^2 \cap C$ satisfies $\alpha \cdot k = d$ and $k \in B_{R\sqrt{d}}$. Since

$$x_i = \frac{\alpha_i \cdot (k - x^*)}{\sqrt{dp_i q_i}}$$

the assumption that $k \in B_{R\sqrt{d}}$ ensures that the $x_i$ remain bounded as $d \to \infty$. Note that, by Proposition 9, the monomial $\prod_{i=1}^{N} p_i^{k_i}$ depends on $k$ only via $\alpha \cdot k$, and hence here is independent of $k$:

$$\prod_{i=1}^{N} p_i^{k_i} = \prod_{i=1}^{N} p_i^{\alpha_i \cdot x^*} = \prod_{i=1}^{N} p_i^{dp_i}$$

Furthermore

$$\sum_{i=1}^{N} q_i x_i^2 = \frac{(k - x^*)^T A (k - x^*)}{d}$$

where $A$ is the positive-definite $2 \times 2$ matrix given by

$$A = \sum_{i=1}^{N} \frac{1}{p_i} \alpha_i^T \alpha_i$$

Thus, as $d \to \infty$, the ratio

$$\frac{(\alpha \cdot k)!}{\prod_{i=1}^{N}(\alpha_i \cdot k)!} : \frac{\exp\left(-\frac{1}{2d}(k - x^*)^T A (k - x^*)\right)}{(2\pi d)^{\frac{N-1}{2}} \prod_{i=1}^{N} p_i^{dp_i + \frac{1}{2}}} \to 1 \tag{9}$$

for all $k \in \mathbb{Z}^2 \cap C \cap B_{R\sqrt{d}}$ such that $\alpha \cdot k = d$.

**Theorem 10.** (This is Theorem 6 in the main text). Let $X$ be a toric variety of Picard rank two and dimension $N - 2$ with weight matrix

$$\begin{pmatrix} a_1 & a_2 & a_3 & \cdots & a_N \\ b_1 & b_2 & b_3 & \cdots & b_N \end{pmatrix}$$

Let $a = a_1 + \cdots + a_N$ and $b = b_1 + \cdots + b_N$, let $\ell = \gcd\{a, b\}$, and let $[\mu : \nu] \in \mathbb{P}^1$ be the unique solution to Eq. (8). Let $c_d$ denote the coefficient of $t^d$ in the regularised quantum period $\widehat{G}_X(t)$. Then non-zero coefficients $c_d$ satisfy

$$\log c_d \sim Ad - \frac{\dim X}{2} \log d + B$$

as $d \to \infty$, where

$$A = -\sum_{i=1}^{N} p_i \log p_i$$

$$B = -\frac{\dim X}{2} \log(2\pi) - \frac{1}{2} \sum_{i=1}^{N} \log p_i - \frac{1}{2} \log\left(\sum_{i=1}^{N} \frac{(a_i b - b_i a)^2}{\ell^2 p_i}\right)$$

and $p_i = \frac{\mu a_i + \nu b_i}{\mu a + \nu b}$.

**Proof.** We need to estimate

$$c_d = \sum_{\substack{k \in \mathbb{Z}^2 \cap C \\ \text{with } \alpha \cdot k = d}} \frac{(\alpha \cdot k)!}{\prod_{i=1}^{N}(\alpha_i \cdot k)!}$$

Consider first the summands with $k \in \mathbb{Z}^2 \cap C$ such that $\alpha \cdot k = d$ and $k \notin B_{R\sqrt{d}}$. For $d$ sufficiently large, each such summand is bounded by

$cd^{-\frac{1+\dim X}{2}}$ for some constant $c$—see Eq. (9). Since the number of such summands grows linearly with $d$, in the limit $d \to \infty$ the contribution to $c_d$ from $k \notin B_{R\sqrt{d}}$ vanishes.

As $d \to \infty$, therefore

$$c_d \sim \frac{1}{(2\pi d)^{\frac{N-1}{2}} \prod_{i=1}^N p_i^{dp_i+\frac{1}{2}}} \sum_{\substack{k \in \mathbb{Z}^2 \cap C \cap B_{R\sqrt{d}} \\ \text{with } k \cdot d = d}} \exp\left(-\frac{(k-x^*)^T A (k-x^*)}{2d}\right)$$

Writing $y_k = (k - x^*)/\sqrt{d}$, considering the sum here as a Riemann sum, and letting $R \to \infty$, we see that

$$c_d \sim \frac{1}{(2\pi d)^{\frac{N-1}{2}} \prod_{i=1}^N p_i^{dp_i+\frac{1}{2}}} \sqrt{d} \int_{L_\alpha} \exp\left(-\frac{1}{2} y^T A y\right) dy$$

where $L_\alpha$ is the line through the origin given by $\ker \alpha$ and $dy$ is the measure on $L_\alpha$ given by the integer lattice $\mathbb{Z}^2 \cap L_\alpha \subset L_\alpha$.

To evaluate the integral, let

$$\alpha^\perp = \frac{1}{\ell}\begin{pmatrix} b \\ -a \end{pmatrix} \quad \text{where } \ell = \gcd\{a,b\}$$

and observe that the pullback of $dy$ along the map $\mathbb{R} \to L_\alpha$ given by $t \mapsto t\alpha^\perp$ is the standard measure on $\mathbb{R}$. Thus

$$\int_{L_\alpha} \exp\left(-\frac{1}{2} y^T A y\right) dy = \int_{-\infty}^{\infty} \exp\left(-\frac{1}{2}\theta x^2\right) dx = \sqrt{\frac{2\pi}{\theta}}$$

where $\theta = \sum_{i=1}^N \frac{1}{\ell^2 p_i}(\alpha_i \cdot \alpha^\perp)^2$, and

$$c_d \sim \frac{1}{(2\pi d)^{\frac{\dim X}{2}} \prod_{i=1}^N p_i^{dp_i+\frac{1}{2}} \sqrt{\theta}}$$

Taking logarithms gives the result. □

## Data availability
Our datasets[42,43] and the code for the Magma computer algebra system[44] that was used to generate them are available from Zenodo[45] under a CC0 license. The data was collected using Magma V2.25-4.

## Code availability
All code required to replicate the results in this paper is available from Bitbucket under an MIT license[46].

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

## Acknowledgements

T.C. is funded by ERC Consolidator Grant 682603 and EPSRC Programme Grant EP/N03189X/1. A.M.K. is funded by EPSRC Fellowship EP/N022513/1. S.V. is funded by the EPSRC Centre for Doctoral Training in Geometry and Number Theory at the Interface, grant number EP/L015234/1. We thank Giuseppe Pitton for conversations and experiments that began this project, and thank John Aston and Louis Christie for insightful conversations and feedback.

## Author contributions

T.C., A.M.K., and S.V. contributed equally to this work.

## Competing interests

The authors declare no competing interests.
