## [Peer Review File · Nature Communications]

Machine Learning the Dimension of a Fano VarietyREVIEWER COMMENTS

Reviewer #1 (Remarks to the Author):

The paper is very nicely written with a detailed description of the data generation process. The results and conclusions are presented clearly using appropriate graphs. It is also nice to see rigorous analysis being done following data exploration.

The title however suggests that the focus of the paper is on machine learning whereas in the paper very little focus is given to this. The neural networks section is very short and does not seem so necessary since the same conclusions can be made from the data analysis and later analysis sections (without Machine-learning).

Most importantly, the topic of this paper seems too technical for Nature Comm and might be better suited to a specialized journal.

Reviewer #2 (Remarks to the Author):

The paper generated data of the regularized quantum periods and dimensions of toric varieties with Picard rank ≤ 2 and with at worst terminal singularities. It then fit the data into machine-learning models for computing the dimensions from quantum periods.

Moreover, the paper constructed a formula relating asymptotics of coefficients of the quantum periods and the dimensions for toric varieties with Picard rank ≤ 2 . It is a simple application of a formula of quantum periods in their earlier paper 'Quantum periods for 3-dimensional Fano manifolds'.

The machine learning techniques employed in the paper are classical and not new. On the other hand, it shows an interesting instance of application of methods in data science to study topics in pure mathematics. I recommend acceptance of the paper.

Reviewer #3 (Remarks to the Author):

This article utilizes machine learning (ML) to determine the dimension of a Fano variety. The datasets consist of the regularized quantum periods of Fano varieties, and the ML algorithms adopted here are standard ones such as support vector machine, random forest and neural networks. An ML approach to mathematical structures is emerging as a paradigm shift in mathematics research and this paper successfully provides another supporting evidence for the potential of this new trend. After obtaining high accuracy in classification, the authors formulate rigorous statements to asymptotically determine the dimension of a Fano variety and hence strengthen the expectation that the regularized quantum period characterizes a Fano variety uniquely. I found that this article suggests an exemplary way to follow in mathematics research using ML methods.

I have questions that I hope to be answered in a revision of this paper.

1. How much do weighted projective spaces and toric varieties of Picard rank two cover the set of Fano varieties? Do they form very special families of Fano varieties? Can you apply ML techniques for other classes of Fano varieties?
2. How much crucial was your ML approach in formulating Theorems 5 and 6? If it could have been difficult without ML, that fact should be explained and emphasized.
3. Your use of slope and y-intercept is reasonable. However, what happens if you instead use the first two principal components of PCA?
4. You exclude the first 999 d's to reduce noise in linear regression, while you include the first 100 d's to significantly enhance accuracy in MLP_102. That sounds like a contradiction. You vaguely mention that the first 100 carry some information. Can you provide a better explanation?

The following are minor comments that, I think, need to be addressed in a revision.

- a. around lines 96-106: You are using words "terminal quotient singularities", "quotient

singularities" and "terminal singularities". Are they all different or all the same?

b. I was wondering how often $c_d=0$, and later found in Theorem 5 that $c_d=0$ unless d is divisible by a . It would be better if it is clarified earlier. Is this fact reflected in figures, e.g., in Figure 1? It is not clear.

c. The axes in Figures 7 and 9 do not match with the captions. Are they correct?

d. The transition to A and B from slope and y -intercept needs to be explained more clearly.

e. line 10 in Supplement: Mentioning smoothness again sounds redundant.

f. Figure 1 in Supplement: These pictures are real surfaces, but you are explaining complex surfaces. Clarification would be helpful.

Reviewer #1.

The paper is very nicely written with a detailed description of the data generation process. The results and conclusions are presented clearly using appropriate graphs. It is also nice to see rigorous analysis being done following data exploration.

The title however suggests that the focus of the paper is on machine learning whereas in the paper very little focus is given to this. The neural networks section is very short and does not seem so necessary since the same conclusions can be made from the data analysis and later analysis sections (without Machine-learning).

We have revised the exposition to emphasize that our contribution is the use of machine learning for theorem discovery and structure discovery in pure mathematics. Thus although the ML analysis is not used in the final proof of Theorems 5 and 6, we feel that retaining it is important because it was essential to our discovery of the statements. See the revised paragraph at lines 50–65 and the new paragraph at lines 386–396.

Most importantly, the topic of this paper seems too technical for Nature Comm and might be better suited to a specialized journal.

We revised our exposition to explain why the results may be of interest to a broader audience. See the revised paragraphs at lines 7–21 and the new paragraphs at lines 34–37 and 67–81.

Reviewer #2.

The paper generated data of the regularized quantum periods and dimensions of toric varieties with Picard rank ≤ 2 and with at worst terminal singularities. It then fit the data into machine-learning models for computing the dimensions from quantum periods.

Moreover, the paper constructed a formula relating asymptotics of coefficients of the quantum periods and the dimensions for toric varieties with Picard rank ≤ 2 . It is a simple application of a formula of quantum periods in their earlier paper ‘Quantum periods for 3-dimensional Fano manifolds’.

In the new paragraph on lines 386–396 we clarified the relationship between the proof of Theorem 6 and prior mathematical work.

The machine learning techniques employed in the paper are classical and not new. On the other hand, it shows an interesting instance of application of methods in data science to study topics in pure mathematics. I recommend acceptance of the paper.

Reviewer #3.

This article utilizes machine learning (ML) to determine the dimension of a Fano variety. The datasets consist of the regularized quantum periods of Fano varieties, and the ML algorithms adopted here are standard ones such as support vector machine, random forest and neural networks. An ML approach to mathematical structures is emerging as a paradigm shift in mathematics research and this paper successfully provides another supporting evidence for the potential of this new trend. After obtaining high accuracy in classification, the authors formulate rigorous statements to asymptotically determine the dimension of a Fano variety and hence strengthen the expectation that the regularized quantum period characterizes a Fano variety uniquely. I found that this article suggests an exemplary way to follow in mathematics research using ML methods.

I have questions that I hope to be answered in a revision of this paper.

1. How much do weighted projective spaces and toric varieties of Picard rank two cover the set of Fano varieties? Do they form very special families of Fano varieties? Can you apply ML techniques for other classes of Fano varieties?

We added a new section that discusses this in detail. See lines 196–215 of the Supplementary Material.

2. How much crucial was your ML approach in formulating Theorems 5 and 6? If it could have been difficult without ML, that fact should be explained and emphasized.

The use of ML was essential to our discovery of Theorems 5 and 6, and we have revised the exposition to emphasise this. See the revised text at lines 50–65 and the new paragraph at lines 386–396.

3. Your use of slope and y -intercept is reasonable. However, what happens if you instead use the first two principal components of PCA?

The first two components of PCA seem to carry the same information as the slope and intercept, and we added a new section (on lines 181–195 of the Supplementary Material) that discusses this. Because the patterns of zeroes in the period sequences (c_d) differ, and we analyse $(\log c_d)$, it only makes sense to run PCA analysis on Fano varieties of a fixed Fano index r . For each r , the pair (slope, y -intercept) is related to $(\text{PCA}_1, \text{PCA}_2)$ by an invertible affine-linear transformation, but this transformation varies with r .

In thinking about this suggestion, we decided to go back and perform additional experiments using components of PCA as features. We expected that these experiments would produce negative results, based on our understanding of the mathematics and our experience of analysing similar questions at the start of the project, and indeed this was the case. We would therefore prefer not to include the experiments in our paper, but describe them here. The code for these experiments is included in the supporting material as the Jupyter notebook `revision_notebook.ipynb`.

- (I) The first experiment uses all 100 000 terms of the sequence (c_d) for the weighted projective spaces dataset, taking log of the non-zero terms and leaving the zero terms unchanged. This approach was not very successful, and was very computationally expensive (as we have to perform partial fits for the PCA because the data does not fit in memory). The PCA explained variance ratios for the first two components are (0.014, 0.011), and when we plot PCA_1 against PCA_2 there is no clustering behaviour exhibited. See Figure 1a below. We trained a Random Forest Classifier for the dimension on the features $(\text{PCA}_1, \text{PCA}_2)$ and obtained 53% accuracy.

- (II) The second experiment uses the first 100 terms of the sequence (c_d) for the Picard rank two dataset, taking log of the non-zero terms and leaving the zero terms unchanged. This approach is again not very successful: the PCA explained variance ratios are $(0.148, 0.053)$, and when we plot PCA_1 against PCA_2 there is no clustering behaviour. See Figure 1b below.
- (III) The third experiment uses $(\log c_d)$ for the weighted projective spaces dataset, where in each case d ranges over the sequence such that $c_d \neq 0$. Depending on the weights, the number of non-zero c_d varies, so in order to fit the PCA we consider only the first m non-zero terms of each, where m is the minimal number of non-zero terms encountered. Without knowing the statement of Theorem 5 it is unclear why this would be a sensible approach, since *a priori* we appear to be comparing terms c_d for different values of d . We therefore had not taken this approach before. This experiment was very successful, with PCA explained variance ratios $(9.99999998 \times 10^{-1}, 2.16403642 \times 10^{-9})$ and clustering behaviour, obtained when plotting PCA_1 against PCA_2 , revealing a ‘stretched out’ version of Figure 3a in the main text. See Figure 1c below.
- (IV) The fourth experiment uses $(\log c_d)$ for the Picard rank two dataset. Again, depending on the weights, the number of non-zero c_d varies, so in order to fit the PCA we consider only the first m non-zero terms of each, where m is the minimal number of non-zero terms encountered. Without knowing the statement of Theorem 6 it is unclear why this would be meaningful, for the same reason as above, and so we had not taken this approach before. This experiment was more successful than experiment (II), with PCA explained variance ratios $(0.993, 0.004)$ and clustering behaviour, obtained when plotting PCA_1 against PCA_2 , somewhat reminiscent of Figure 3b in the main text. See Figure 1d below.

We trained a Random Forest Classifier for the dimension on the features $(\text{PCA}_1, \text{PCA}_2)$ and obtained 78.8% accuracy. It is not fair to compare this with the RFC model in the main text, which is trained on a smaller dataset that excludes samples with high standard error. Retraining the RFC from the main text on the entire dataset, so that the results are comparable, gave 85% accuracy.

Experiments (III) and (IV) suggest that doing PCA on the vector

$$(\log c_d)_{d \text{ such that } c_d \neq 0}$$

is a reasonable approach to dimensionality reduction in our setting. However, features extracted from the linear regression have a clearer mathematical interpretation and lead to more accurate models.

Experiment (II) here is slightly unsatisfactory, because it involves only 100 terms of the period sequence, but it was the best that we could do without generating additional period sequence data. Given how unsuccessful experiment (I) was, and the computational costs involved, we chose not to pursue this line further.

4. You exclude the first 999 d 's to reduce noise in linear regression, while you include the first 100 d 's to significantly enhance accuracy in MLP_{102} . That sounds like a contradiction. You vaguely mention that the first 100 carry some information. Can you provide a better explanation?

Our exposition was unclear here and we have improved it. See the new text on lines 271–274 of the main text and the new paragraph on lines 156–168 of the Supplementary Material. We also performed a new experiment, reported in the Supplementary Material: omitting the slope and intercept from the feature set and training an MLP classifier for dimension on the coefficients $\log c_d$ for $1 \leq d \leq 100$ gave an accuracy of only 62%.

FIGURE 1. (A) Clustering analysis for PCA components in the weighted projective space case, from experiment (I) above. (B) Clustering analysis for PCA components in the Picard rank two case, from experiment (II) above. (C) Clustering analysis for PCA components in the weighted projective space case, from experiment (III) above. (D) Clustering analysis for PCA components in the Picard rank two case, from experiment (IV) above.

The following are minor comments that, I think, need to be addressed in a revision.

a. around lines 96-106: You are using words “terminal quotient singularities”, “quotient singularities” and “terminal singularities”. Are they all different or all the same?

Thank you for insisting on clarity here. We have revised the exposition to remove this confusion and now have a single notion, of terminal quotient singularities, that is used throughout the document. We introduce and motivate this in the new discussion on lines 75–81, and it also leads to many small changes in wording later in the paper: on lines 118, 123–125, 127–128, 134, 136, 141, 155, 168, 180–183, 266, 369–370, and in the captions to Figures 3, 6, and 7. It also leads to small expository changes in the Supplementary Material: on lines 63–67 and 71–72, in the caption to Supplementary Table 1, and in the captions to Supplementary Figures 2, 3, and 4 (which was previously Supplementary Figure 5).

b. I was wondering how often $c_d = 0$, and later found in Theorem 5 that $c_d = 0$ unless d is divisible by a . It would be better if it is clarified earlier. Is this fact reflected in figures, e.g., in Figure 1? It is not clear.

We now highlight this on lines 119–122, and replotted Figure 2 (which was formerly Figure 1) and also Supplementary Figure 5 (which was formerly Supplementary Figure 4) in the Supplementary Material to make this point clearer.

c. The axes in Figures 7 and 9 do not match with the captions. Are they correct?

This was a mistake. We have corrected the axis labels. (The Figures are now Figures 7a and 7b.)

d. The transition to A and B from slope and y -intercept needs to be explained more clearly.

We improved the exposition here, adding new text on lines 325–340 and a new figure as Figure 8. We also corrected a typo in the formula relating B to the slope of the linear model.

e. line 10 in Supplement: Mentioning smoothness again sounds redundant.

We revised this.

f. Figure 1 in Supplement: These pictures are real surfaces, but you are explaining complex surfaces. Clarification would be helpful.

We moved the discussion of this to a new section in the main text at the start of the Results: see lines 67–74. We also changed the captions to Supplementary Figure 1 in the Supplementary Material to emphasise this point.

Additional changes.

In the main text, we changed the section titles, table captions, figure captions, references to the Supplementary Material, bibliography, and colour gradients in images as requested in the formatting instructions. We also moved the data availability and code availability statements to the correct place, which caused changes on lines 157, 185, and 466–471, and made minor expository improvements as follows:

- stylistic changes on lines 22, 29, 48–49, 86, and 187;
- included an omitted word on line 44;
- removed some unnecessary text on line 64;
- reordered the discussion in the ‘Theoretical analysis’ subsection: see lines 341–348, 350, and 361.

In the Supplementary Material we changed the section titles, table captions, table references, figure captions, figure references, bibliography, and colour gradients in images as requested in the formatting instructions. We also deleted some unnecessary text on line 91, added an additional subtitle on line 169, and corrected a typographical error in the accuracy for MLP_2 on line 143 and in Supplementary Table 3. Moreover, we fixed the axes in the confusion matrices in Supplementary Figures 10, 12, and in Supplementary Table 4, which were originally swapped by mistake.

REVIEWERS' COMMENTS

Reviewer #2 (Remarks to the Author):

The paper has been nicely revised. I recommend acceptance of the paper.

Reviewer #3 (Remarks to the Author):

I read the revised version and found that all my concerns and comments had been taken care of satisfactorily and thoroughly. I have no additional comments and would recommend this version for publication through Nature Communications.

Reviewer #2.

The paper has been nicely revised. I recommend acceptance of the paper.

We thank the referee for their time and their insightful comments.

Reviewer #3.

I read the revised version and found that all my concerns and comments had been taken care of satisfactorily and thoroughly. I have no additional comments and would recommend this version for publication through Nature Communications.

We thank the referee for their time and their insightful comments.